# Variation in Wheat Quality and Starch Structure under Granary Conditions during Long-Term Storage

**DOI:** 10.3390/foods12091886

**Published:** 2023-05-04

**Authors:** Hao Hu, Mingming Qiu, Zhuzhu Qiu, Shipeng Li, Lintao Lan, Xingquan Liu

**Affiliations:** 1College of Food and Health, Zhejiang Agriculture and Forest University, Hangzhou 311300, China; 20180015@zafu.edu.cn (H.H.);; 2Food and Strategic Reserves Bureau of Quzhou City, Quzhou 324199, China

**Keywords:** wheat quality, physicochemical indicator, starch structure

## Abstract

As a globally distributed cereal, wheat is an essential part of the daily human dietary structure. Various changes in nutrient composition and starch structure can reflect the quality of wheat. In this study, we carried out a series of measurements to reveal the levels of wheat quality during long-term storage. We found that the deterioration of wheat was apparent after two years of storage: (1) the content of fatty acid increased from 12.47% to 29.02%; (2) the malondialdehyde content increased to 37.46%; (3) the conductivity significantly increased from 35.71% to 46.79%; and (4) other indexes, such as the amylopectin content, peak viscosity, and disintegration rate, increased noticeably during storage. Moreover, SEM images revealed a certain degree of damage on the surface of starch granules, and an X-ray diffraction (XRD) analysis showed A-type crystalline starch of wheat. Additionally, FTIR spectra suggested that the ratio of amylose and amylopectin decreased with a decreasing content of amylose and increasing content of amylopectin. The ratio of amylose and amylopectin can lead to variations in wheat machining characteristics. Therefore, wheat should be kept at an average of 20 °C with safe water content for less than two years to maintain reasonable quality.

## 1. Introduction

Wheat refers to several species of the genus *Triticum* and their edible grains, and they are a staple food that is part of dietary structure globally [1,2]. Since the outbreak of the coronavirus pandemic, major grain-exporting countries, such as the USA, Canada, Australia, Russia, and Ukraine, restricted food exportation, thus triggering market tensions. Moreover, the Russo-Ukrainian War exacerbated this situation, leading to a further restriction of wheat supply. Therefore, the technique of wheat reserve is becoming indispensable to maintain its high quality. The duration of storage greatly influences the physicochemical properties and nutrient content of wheat, in addition to the environmental factors that are commonly reported [3]. The substances of newly harvested wheat undergo a series of spontaneous changes in biochemical metabolism, further affecting the nutrition and quality of wheat. As storage duration is extended, wheat experiences severe aging and even deterioration, such as a significant increase in fatty acid and malondialdehyde content and a decrease in the germination rate [4].

Starch is the main component of wheat, and its content is closely related to wheat quality [5]. Its main molecular types are amylose and amylopectin, of which amylose is composed of α-D-(1,4)-glucose units with a nearly linear structure, while amylopectin comprises the main α-D-(1,4)-glucose chains linked by α-D-(1,6)-glucose bonds [6,7]. Their ratio contributes to the starch structure and subsequently influences the processing property of wheat [8]. Starch and protein are the main components of wheat, and they can influence its processing properties: the pasting properties, thermal properties, gel consistency, and swelling potential of starch in wheat will affect the quality of wheat processed into flour products, i.e., bread, noodles, steamed bread, etc. [9]; the content and composition of protein affect the rheological and baking properties of dough [10]. Higher pasting peak viscosity, breakdown, and swelling value will improve the quality of noodles. Other key factors such as morphology, crystal structure, and lamellar structure lead to variations in starch structure. Apart from that, phenolic compounds are quite important for the nutrition of wheat-based foods [11]. Phenolic substances mainly include phenolic acids and flavonoids, and their distribution is mainly concentrated in wheat bran. Flavonoids are an important natural antioxidant that can effectively scavenge excessive free radicals to prevent damage to proteins and DNA in organisms [12]. In addition, when the flavonoid content in the gut reaches a certain level, it can help the human body to eliminate free radicals and chelate redox-active metals, thus preventing cells from oxidative stress and inhibiting organismal aging. However, the content of flavonoids decreases significantly with prolonged storage time, resulting in a decrease in the nutritional value and anti-oxygenation of wheat [13]. Furthermore, flavonoids have been found to modulate the intestinal immune function both directly and indirectly, as it can be metabolized by enzymes in intestinal epithelial cells and intestinal microflora. Other investigations have found that flavonoids can interact with the gastrointestinal microflora to maintain the immune homeostasis in the gastrointestinal tract (GIT) and further facilitate nutrient absorption [1,12]. Hence, it is essential to investigate variations in wheat in terms of physicochemical properties, nutrients, and starch structure during long-term storage.

Although some previous studies have focused on changes in quality and nutrient content during storage, most of them were carried out under laboratory conditions. How they vary under granary conditions and whether the results are consistent with those in the laboratory also deserve to be studied. To address this issue, we conducted a long-term follow-up survey to investigate changes in the physicochemical properties and nutritional compositions of wheat during different storage periods. The findings of this study can provide essential insights into maintaining optimal wheat quality during practical storage.

## 2. Materials and Methods

### 2.1. Materials

The samples were durum wheat from Australia, stored for 0, 1, and 2 years at the Quzhou grain depot in Zhejiang Province at 20–25 °C in summer and 10–15 °C in other seasons. The wheat stored in different years was subject to different treatments. Each treatment was composed of three samples from three different silos, and a total of nine samples were used in this study.

### 2.2. Wheat Quality Variation in Storage

#### 2.2.1. Determination of Water Content

Water content was measured based on the method reported by Zhai et al. [4]. An aluminum weighing box was placed in a 105 °C dryer for drying. After cooling, the mass of the empty box was weighed. Wheat grinding powders (80-mesh sieved) with different storage times were weighed, and then, about 2 g of each was placed in the weighing box; the powders were dried again and then weighed.

#### 2.2.2. Determination of Fatty Acid Value

The fatty acid value was determined following the method proposed by Zhai et al. [4] after the wheat samples’ pretreatment, which were shelled, milled, shaken, and filtered to make a filtrate. Then, 10 mL of filtrate was transferred into a conical flask, after which, 20 mL of ultra-pure water with 3–4 drops of phenolphthalein solution was added. Then, the solution was titrated with KOH until it appeared light pink for 3 s. Ethanol solution (10 mL) was set as control.

#### 2.2.3. Determination of Protein Content

The contents of glutenin, gliadin, globulin, and albumin were determined by the method reported by Zhao et al. with slight modifications [14].

#### 2.2.4. Determination of Malondialdehyde Content

According to the study reported by Qu et al. [15], samples and 10 mL of trichloroacetic acid (TCA) were added into iodine bottles, and then shaken and placed on a 50 °C thermostatic oscillator, followed by centrifugation at 4000 rpm for 10 min. Afterward, the supernatant and standard series of solutions were placed into the plug tube. In addition, 5 mL of TCA was removed as control, and then 5 mL of thiobarbituric acid (TBA) was added. After mixing, the solution was cooled in a water bath, and the absorbances of the standard solutions and sample solutions were recorded.

#### 2.2.5. Determination of Germination Rate

The germination rate was investigated on the basis of the reported method [16]. The sterilized filter paper was wetted with sterilized water to reach saturation in plastic boxes. Each set of 100 intact seeds was selected from wheat stored for different years. Then, they were placed in the incubator for germination. Sterilized water was added every three days to ensure suitable moisture for germination. The number of germinated seeds was counted after 7 days. 

#### 2.2.6. Determination of Electric Conductivity

According to the method mentioned by Qu et al. [15], 40 intact wheat seeds with different storage years were selected and divided into 4 groups of 10 seeds each, followed by being weighed and added to test tubes. After being soaked with sterilized water (50 mL) for 13 h, the conductivity of the sample soaking solution was measured at room temperature. 

### 2.3. Wheat Starch Extraction

Wheat was peeled with a ratio of 1:5 soaked in distilled water overnight after beating the slurry through 200-mesh sieves, with distilled water used to wash the slurry. The washed suspension was collected in a 50 mL centrifuge tube; then, it was centrifuged at 4000 rpm for 10 min, and the upper layer of protein was scraped until clean. Then, it was dried at 42 °C, and passed through a 100-mesh sieve [17].

### 2.4. Determination of Water Solubility and Swelling Power 

The sample (about 0.1 g) was transferred into a test tube containing 10 mL deionized water (empty tube as M1). Then, it was heated at 90 °C for 60 min. When the sample was cooled to room temperature, it was centrifuged at 8000× *g* for 15 min. Subsequently, the tube containing sediment was dried at 80 °C for 1 h and weighed, and recorded as M3. The tube containing precipitation was dried at 80 °C for 20 min and recorded as M2. The aluminum bucket with supernatant was dried to constant weigh in an oven at 80 °C, then placed in a constant temperature and humidity chamber for 20 min, weighed, and recorded as M4 [18,19]. 

The swelling power (SP) and water solubility (WS) of starch were calculated according to the following formula:$$\mathrm{WS}\;(\%)=\frac{M4-M3}{M}\times 100\%$$
$$\mathrm{SP}\;(g/g)=\frac{M2-M1}{M\times \left(1-\mathrm{WSI}\right)}$$

### 2.5. Determination of Adhesive Strength of Wheat Starch

The samples (100 mg) were placed in a 15 mL glass tube, then 2 mL of 0.2 M potassium hydroxide solution was mixed with them before being immediately placed in a boiling water bath. The boiling starch glue was maintained at about two-thirds of the tube volume and gelatinized for 8 min. After gelatinization, samples were cooled in an ice bath for 20 min. Then, they were placed on the horizontal platform at 25 ± 2 °C for 1 h and the fly-out length of the sample in the tube was measured.

### 2.6. Determination of Thermal Property of Wheat Starch

The starch extracted from wheat was ground, dispersed, and passed through a 100-mesh sieve. An appropriate amount of sample (2.5–3.1 mg) was placed into the sample tray. Then, the starch was equilibrated by adding deionized water at 4 °C for 24 h. The enthalpy variation of the sample was analyzed by differential scanning calorimetry (NETZSCH DSC 200 F3) with the temperature increasing from 30 °C to 105 °C at a speed of 10 °C/min. Thermal transformations were estimated by detecting the change in heat absorption and release of the samples [20].

### 2.7. Determination of Pasting Property of Wheat Starch

The pasting property of wheat starch was determined following the method described by Wu et al. [21]. The viscosity curve and seven characteristic points of starch during the pasting process were obtained by a rapid viscosity analyzer (RVA). The viscosity unit was expressed by ‘cP’.

### 2.8. Determination of Amylose Content and Amylopectin Chain Length Distribution

The apparent amylose content of wheat starch was determined by iodine reagent method and calculated according to the standard curve obtained from different proportions of the amylose and amylopectin mixtures [7]. 

The amylopectin chain length distribution of wheat starch was determined according to the method mentioned by Ren et al. [22]. Purified starch (10 mg) was mixed with isoamylase in 50 μL sodium acetate (0.6 M, pH 4.4). Then, the chain length distribution of starch was detected using a Thermo ICS5000 ion chromatography system (Thermo Fisher Scientific, Waltham, MA, USA).

### 2.9. Wheat Starch Structure Analysis 

Wheat starch structure was analyzed by Fourier transform infrared spectroscopy (FTIR), referring to the method described by Sun et al. [17]. The short-range ordered structure of starch was analyzed by NICOLET iS50FTIR with a resolution of 4 cm^−1^ in the wavelength range of 4000 cm^−1^ to 400 cm^−1^.

### 2.10. Wheat Starch Crystal Structure Analysis 

The crystal structure of wheat starch was analyzed by an X-ray diffractometer (XRD). All samples were scanned using a Cu Kα X-ray source in the step size range of 4–40° and 0.033° [23]. The voltage and current were 40 kV and 40 mA, respectively. The calculation of relative crystallinity (RC) was conducted using Jade software version 6.0 (Material Data Corporation, CA, USA).

### 2.11. Wheat Starch Granule Variation 

The surface of wheat starch granules was observed using scanning electron microscopy (SEM). The analysis was carried out as described by Hung and Morita (2005). A certain amount of starch was suspended in 95% ethanol for a few minutes and then sprinkled onto a double-sided adhesive tape mounted on an aluminum stub. After being dried by vacuum aspiration for several hours, the samples were coated with Pt/Pd and photographed using an SEM apparatus (Hitachi model S-800, Tokyo, Japan) at an accelerating potential of 10 kV [24].

### 2.12. Statistical Analysis

All experiments were carried out in at least three replicates, and the related data are presented as mean value ± standard deviation (SD). Statistical analyses (ANOVA) were conducted by Duncan’s multiple range tests using SPSS 25.0 software. All the figures were plotted using Origin Pro 9.8.0 software.

## 3. Results and Discussion

### 3.1. Variation in Wheat Quality during Storage

To evaluate the wheat quality during long-term storage, we monitored the change in the contents of water, fat acid, protein, and malonaldehyde (MDA), as well as germination rate and conductivity. Water is an essential basis for organism activity and, as such, it can affect the breeding of molds and grain pests [18]. The water content of wheat increased from 10.04% to 11.71%, with the storage time extending to 2 years (Figure 1a), a result which is consistent with a previous study, which showed that the water content of wheat would become high, leading to a decrease in wheat quality, when the storage conditions deteriorated [25]. Throughout storage, wheat lipids undergo gradual hydrolyzation and oxidation, causing the fatty acid content in wheat to increase from 12.47 to 29.02 mg/100 g (Figure 1b). This increase in fatty acid content represents a significant deterioration in wheat quality. In addition, the proteins significantly varied during this process: albumins decreased from 601.59 mg/100 g to 443.88 mg/100 g, while the contents of globulin, gluten, and gliadin gradually increased (Figure 1c). Furthermore, MDA, a final product of lipid oxidation, can combine with proteins and nucleic acids to cause enzyme inactivation and chromosome mutations [9]. The MDA content increased from 8.89% to 11.39% during the two-year storage, indicating that the wheat quality decreased during this process (Figure 1d). The wheat germination rate, an index reflecting grain seed vigor, decreased from 90% to 22% (Figure 1e). Furthermore, electrical conductivity is considered another indicator to evaluate the wheat quality; it increased from 35.37% to 46.79% after storage of 2 years (Figure 1f), because the accumulation of aldehydes and ketones can exacerbate toxicities on the cell membranes and finally result in increased cell membrane permeability and conductivity. In summary, the physicochemical indexes of wheat suggested its quality deterioration during long-term storage.

### 3.2. Water Solubility and Extensibility 

Water solubility (WS) and swelling power (SP) can reflect the interaction between starch and water. With the increase in storage time, the extensibility of wheat starch first decreased to 19.49 g/g, then increased to 23.00 g/g (Table 1). Swelling power is usually affected by the interaction between starch chains in the amorphous and crystalline domains, including the ratio of amylose to amylopectin, their distributions, and the chain length. The highly branched structure of amylopectin can absorb and hold water, which facilitates starch swelling [26]. Moreover, the long-chain amylose can exudate from the starch granules and disperse in the crystalline region during heating, while the short-chain amylopectin fills in the internal network of starch granules to enhance the starch stability and its swelling power [27]. Our results showed that the swelling power of wheat starch decreases with a decrease in amylopectin content. Meanwhile, the water solubility also decreased from 26.90% to 18.92% due to a rearrangement of the starch granules, improving the interaction between the starch chains, thus promoting the aggregation of amylopectin and the amylose–lipid mixture [19].

### 3.3. Gel Consistency

The gel consistency of wheat is relevant to starch gelatinization, and it is affected by the amylose content, that is, a higher amylose content creates lower gel consistency. The gel consistency increased from 121.67 mm to 134.67 mm with the extension of storage time (Table 1) due to the entanglement of molecular chains bound to hydrogen bonds in starch [28]. During wheat starch retrogradation, amylose, as a linear polymer, produces entangled cross-linking through hydrogen bonds to form a three-dimensional structure. These structures then accumulate to form crystals. In contrast, amylopectin has a dendritic branch structure with large intermolecular steric hindrance. As a result, the intermolecular movement or rearrangement through hydrogen bonds is greatly hindered, and the retrogradation rate is slowed down. The decreased amylose content inhibits gel formation during the retrogradation process, which leads to a higher adhesive strength, as shown by the results.

### 3.4. Thermal Property of Wheat Starch

DSC was carried out to investigate the thermal property of wheat starch. As the initial temperature of starch gelatinization increased continuously, the termination temperature does not show a significant change, possibly due to the incomplete accumulation of the crystal structure caused by the degradation of amorphous starch during storage (Table 2, Figure 2a). The thermal property of wheat starch is related to the enthalpy of amylopectin [21], which is consistent with the results in the process of starch crystal melting, while increased amylose content can decrease starch enthalpy and increase the thermal property. In addition, the thermal property can be reflected by the double-helix structure in starch granules that were formed by the cross-action of amylopectin. Therefore, when the amylose content increases with decreased amylopectin, the thermal property will deteriorate [20].

### 3.5. Pasting Property of Wheat Starch

The peak viscosity, the retrogradation value, the disintegration value, and the final viscosity of wheat starch all first decreased and then increased (Figure 2b, Table 3). The peak viscosity increases with the reduced amylose content, which reflects the hydration and expansion of starch granules [29]. The starch shearing resistance accordingly increases with the decrease in amylose content, since amylopectin molecules can interweave with each other, thus improving the peak viscosity of the starch samples [6,30], which is consistent with the above phenomenon. In addition, the disintegration value is utilized to evaluate the starch disintegration degree [29]. The disintegration value increased from 367.3 to 520.3 after 2 years, indicating that the shearing resistance of starch granules decreases. 

The retrogradation value indicates the extent to which the starch molecules recombine with each other through hydrogen bonding during the cooling process, which includes both short- and long-term rebirth. Short-term retrogradation is mainly due to the recrystallization of amylose, while long-term retrogradation is attributed to the recrystallization of amylopectin. Generally, the higher the amylose content, the easier it will be for the retrogradation to occur. However, the amylose content decreased as the retrogradation value increased (Table 3). The retrogradation process is affected by the content of both amylose and protein. Xijun et al. reported that the pure starch mixed with proteins (albumin, globulin, gliadin, and gluten) in different proportions had different retrogradation values [31]. The contents of gliadin and globulin increased during storage (Figure 1c), which is consistent with the retrogradation value variation. This is because gliadin and globulin are part of the methyl group, which contributes to the arrangement of amylose [32]. In addition, the increased water content can promote amylopectin recrystallization [33,34]. When the water content increases, the amylose and amylopectin chains rearrange to form crystals and then accelerate the starch retrogradation, which is consistent with the results in this study [33].

### 3.6. Amylose Content

The content of amylose initially decreased to 24.15% and then increased to 24.42% during the two-year storage (Table 4).

### 3.7. Wheat Starch Structure Variation

X-ray diffraction (XRD) spectra showed that the starch obtained from the wheat with different storage years exhibits similar diffraction intensity at 15°, 17°, 18°, 20°, and 23° (Figure 3a, Table 5). Among them, the single-state diffraction peaks can be found at 15° and 23°, and double-state diffraction peaks appear at 17° and 18°, corresponding to the structural characteristics of A-type crystals. Moreover, a pronounced diffraction peak occurs at 20°, belonging to the structural characteristics of V-type crystals. These results indicated that storage cannot significantly affect the crystal structure of wheat starch. As the storage time increased, the content of the A-type crystal first decreased and then increased, while the B-type crystal exhibited a continuous decrease. Long chains have a tendency to form B-type crystallinity, and hence, its variation can be reflected by the chain length distribution of starch and vice versa. Based on the results, the relative crystallinity did not have a significant variation with the storage time. 

Furthermore, we performed FTIR analysis to investigate the structure of wheat starch. The two typical absorption peaks at 1045 cm^−1^ and 1022 cm^−^^1^ are usually used to evaluate the crystallinity of starch, which are related to the crystalline region and the amorphous region, respectively. Moreover, the ratio of peak intensities at 1045 cm^−^^1^ and 1022 cm^−^^1^ represents the short-range order of starch crystallinity, whereas the ratio of peak intensities at 995 cm^−^^1^ and 1022 cm^−^^1^ reflects the degree of the double-helix structure in starch granules. The ratio between peak intensities at 1045 cm^−^^1^ and 1022 cm^−^^1^ and the ratio between peak intensities at 995 cm^−^^1^ and 1022 cm^−^^1^ for samples with different storage years indicated that there is no new peak during wheat storage, further suggesting that no covalent bonds are formed during this process (Figure 3b). Clearly, the ratio between peak intensities at 1045 cm^−^^1^ and 1022 cm^−^^1^ increased in the first year but decreased after 2 years, which indicates that the degree of order in starch granules first increases and then decreases. The reduced content of amylose leads to the formation of a double helix structure between the branched chains of amylopectin molecules, causing an increase in the order of starch granules. In addition, the fluctuation of helicity tends to be inversely proportional to the amylose content, and the helicity is also negatively correlated with the amylose content [35].

### 3.8. Morphological Structure Variation of Starch

Natural wheat starch contains two types of starch granules, A-type starch granules with a flat or round-cake shape, and B-type starch granules, which are small and spherical in shape [36]. Interestingly, B-type starch granules contain a few A-type starch granules in some cases. The quantity of small granules and damaged granules gradually increases with storage time and there are more A-type granules than B-type granules (Figure 4), which is consistent with the phenomenon described by Yan et al. [37]. Moreover, B-type granules typically have a higher content of amylopectin. Hence, the decreased A-type and increased B-type granules indicated a decreased amylose content. As storage time extended, the surface of starch granules appeared depressed and spoiled, and then the surface gradually became rough due to biochemical reactions, especially the hydrolysis of enzymes.

## 4. Conclusions

In summary, the quality of wheat significantly deteriorated during long-term storage under suitable granary conditions. Specifically, the components, such as fatty acid, protein, and MDA, as well as germination rate and electrical conductivity, clearly varied. Moreover, other characteristics, i.e., swelling property, adhesive strength, and thermal and gelatinization properties, change with storage life, which has a significant impact on the processing characteristics of wheat. Furthermore, as a main component of wheat, the starch structure is transformed, with the ratio of amylose to amylopectin decreasing to 32.31%, and the spoilage of starch granules appears. Therefore, based on the above results, wheat should be stored for less than one year to maintain a reasonable quality.

## Figures and Tables

**Figure 1 foods-12-01886-f001:**
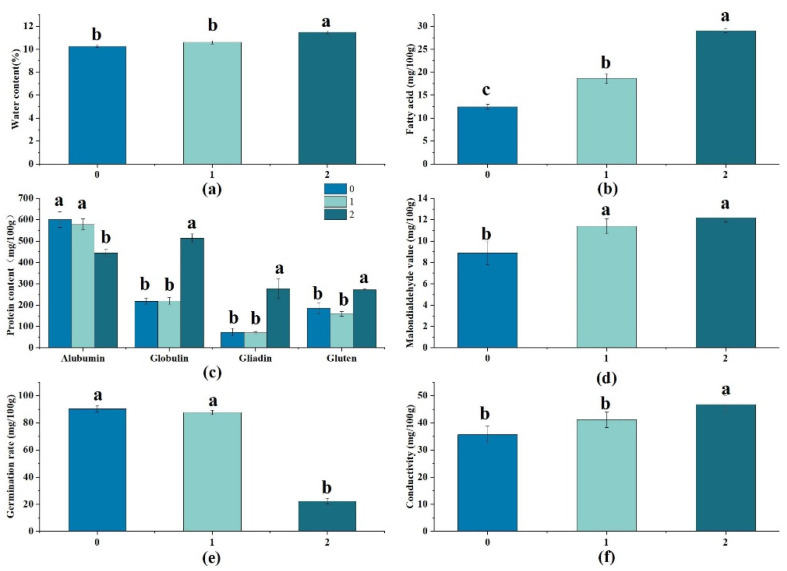
The variation in wheat quality indexes during storage. (**a**) Water content; (**b**) fatty acid value; (**c**) protein content; (**d**) malondialdehyde content; (**e**) germination rate; (**f**) electric conductivity. The numbers 0, 1, and 2 represent different storage years of wheat. The letters of a, b, c indicate the numerical value from high to low.

**Figure 2 foods-12-01886-f002:**
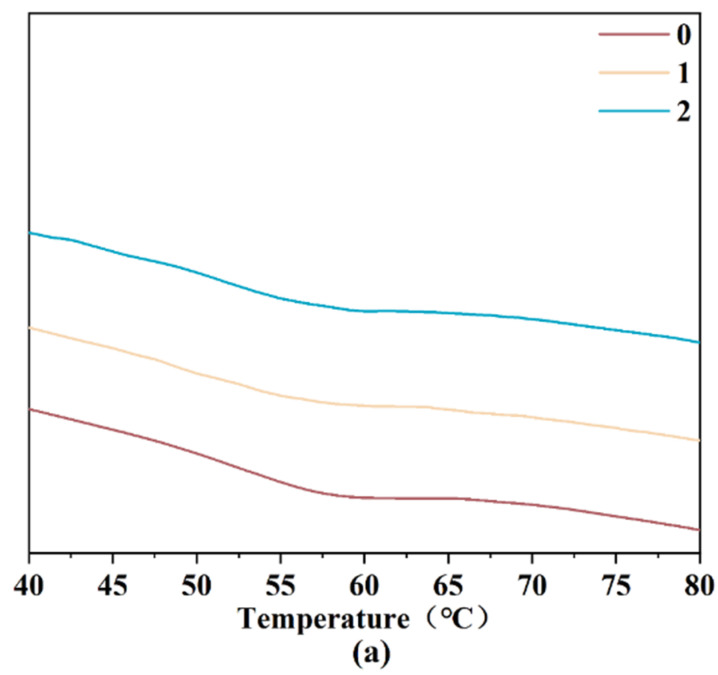
Processing characteristic variations of wheat. (**a**) Wheat starch DSC diagram; (**b**) wheat starch RVA diagram.

**Figure 3 foods-12-01886-f003:**
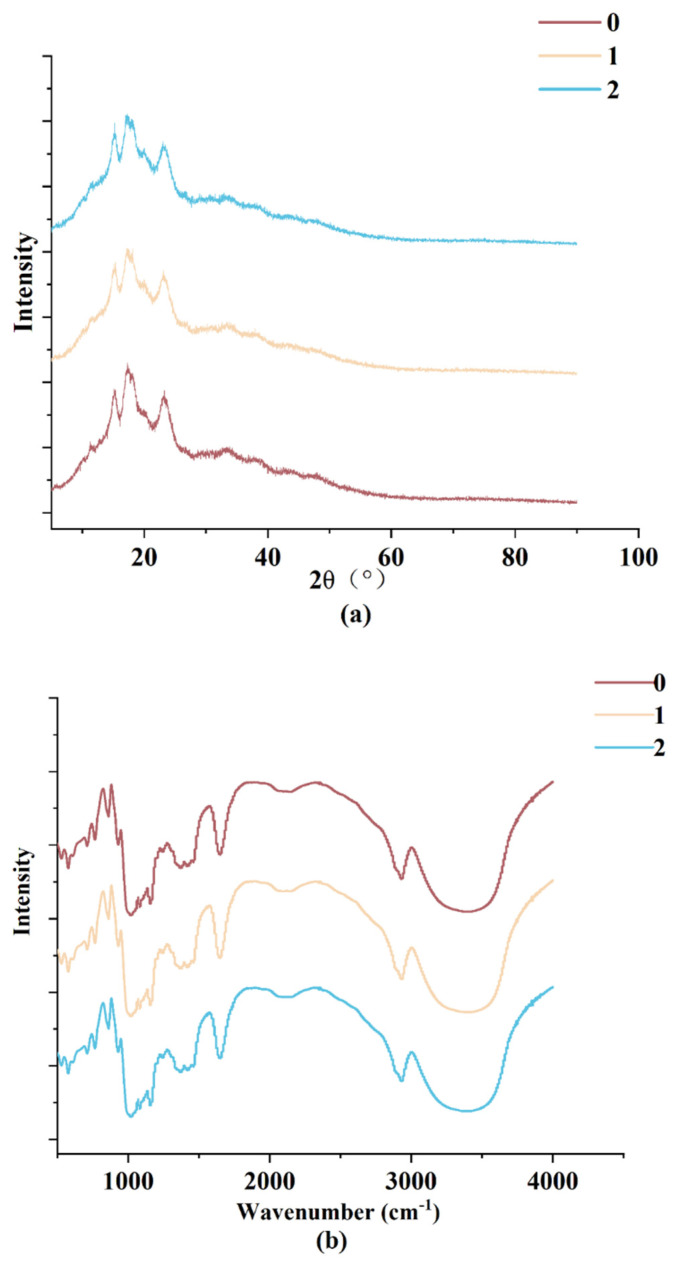
The structure of wheat starch during storage. (**a**) X-ray diffraction; (**b**) FTIR spectra.

**Figure 4 foods-12-01886-f004:**
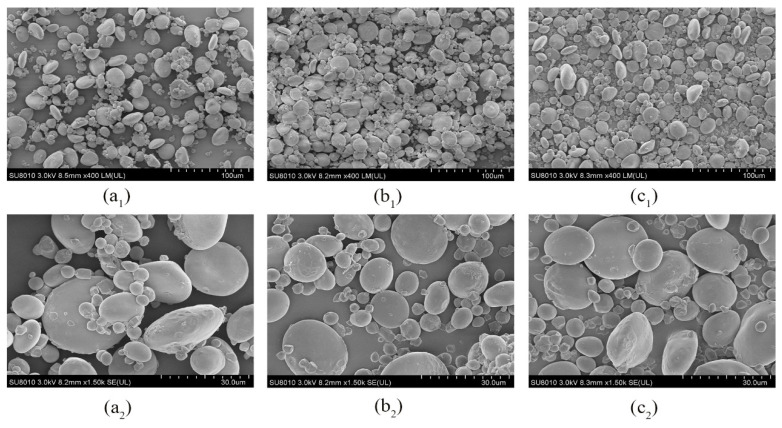
SEM images of starch isolated from wheat. (**a_1_**) 0-year wheat starch with 400× magnifications; (**b_1_**) 1-year wheat starch with 400× magnifications; (**c_1_**) 2-year wheat starch with 400× magnifications; (**a_2_**) 0-year wheat starch with 1500× magnifications; (**b_2_**) 1-year wheat starch with 1500× magnifications; (**c_2_**) 2-year wheat starch with 1500× magnifications.

**Table 1 foods-12-01886-t001:** The water solubility and swelling power variation of wheat starch from different storage times.

Storage Time	Swelling Power (g/g)	Solubility (%)	Flow Length (mm)
0	21.60 ± 0.7420 ab	26.90 ± 1.9370 a	121.67 ± 3.3200 a
1	19.49 ± 0.6845 a	26.69 ± 0.6808 a	130.67 ± 0.8800 b
2	23.00 ± 0.2965 b	18.92 ± 0.2154 b	134.67 ± 2.2700 b

Note: The values are expressed as mean ± SD, which is calculated from duplicated measurement. Values with different letters in the same column mean significantly different (*p* < 0.05).

**Table 2 foods-12-01886-t002:** Pasting property variation of wheat starch.

Storage Time	T_0_ (°C)	Tp (°C)	Tc (°C)	∆H (Jg-1)
0	55.56 ± 0.0424 a	60.04 ± 0.3143 a	64.93 ± 0.0801	2.39 ± 0.0361
1	56.24 ± 0.0660 b	60.58 ± 0.1194 ab	65.13 ± 0.3158	2.50 ± 0.0817
2	57.43 ± 0.0770 c	61.5 ± 0.1744 b	65.32 ± 0.0864	2.34 ± 0.0786

Note: The values are expressed as mean ± SD, which is calculated from duplicate measurements. Values with different letters in the same column mean significantly different (*p* < 0.05).

**Table 3 foods-12-01886-t003:** The pasting parameters of wheat starch from different storage times.

Storage Time	PV (mPa·s)	TV (mPa·s)	BD (mPa·s)	FV (mPa·s)	SB (mPa·s)
0	1039.0 ± 7.8 b	671.7 ± 6.3 b	367.3 ± 5.0 b	1245.3 ± 11.6 b	573.7 ± 7.4 b
1	1015.7 ± 5.8 b	626.3 ± 6.7 c	389.3 ± 1.7 b	1200.0 ± 15.6 b	573.7 ± 9.3 b
2	1347.0 ± 32.8 a	826.7 ± 18.4 a	520.3 ± 14.8 a	1567.3 ± 27.5 a	740.7 ± 9.2 a

Note: The values are expressed as mean ± SD, which is calculated from duplicate measurements. Values with different letters in the same column mean significantly different (*p* < 0.05). The letters of PV, TV, BD, FV, and SB are the abbreviations of peak viscosity, trough viscosity, breakdown viscosity, final viscosity, and setback viscosity, respectively.

**Table 4 foods-12-01886-t004:** The chain length distribution of wheat amylopectin.

Storage Time	AAC (%)	A (%)	B1 (%)	B2 (%)	B3 (%)	ACL (DP)
0	25.00 ± 0.0858 a	29.32 ± 0.0003 a	47.38 ± 0.0001 ab	13.73 ± 0.0002 a	9.74 ± 0.0019	25.04 ± 0.0004
1	24.15 ± 0.3143 b	29.02 ± 0.0001 b	47.62 ± 0a	13.51 ± 0.0016 a	9.68 ± 0	24.96 ± 0.0004
2	24.42 ± 0.1427 ab	29.06 ± 0.0007 b	47.06 ± 0b	14.27 ± 0.0003 b	9.60 ± 0.0003	25.00 ± 0

Note: The values are expressed as mean ± SD, which is calculated from duplicate measurements. Values with different letters in the same column mean significantly different (*p* < 0.05). AAC means apparent amylose content. A, B1, B2, B3, and average chain length (ACL) are the degrees of polymerization (DP) ranging from 6 to 12, 13 to 24, and 25 to 36, respectively.

**Table 5 foods-12-01886-t005:** The short-range ordered structure and crystallinity of wheat starch.

Storage Time	Relative Crystallinity (%)	995/1022	1047/1022
0	34.72 ± 0.1601	1.2525 ± 0.0035 a	1.3253 ± 0.0031 a
1	34.86 ± 0.2263	1.3063 ± 0.0045 b	1.3665 ± 0.0034 b
2	35.48 ± 0.1308	1.1772 ± 0.0025 c	0.8015 ± 0.0014 c

Note: The values are expressed as mean ± SD, which is calculated from duplicate measurements. Values with different letters in the same column mean significantly different (*p* < 0.05).

## Data Availability

Because of the wheat samples were taken from the government grain depot, the data is unavailable due to the government privacy and security restrictions.

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
