# Peer review of "Variation in Wheat Quality and Starch Structure under Granary Conditions during Long-Term Storage"

_foods, 2023, doi:10.3390/foods12091886_

Round 1
Reviewer 1 Report
The aim of the study is clearly introduced. While some results are well analyzed, in my opinion, the structural characterization of starch is overinterpreted, and changes in starch structure over storage time are barely supported by the data. Additionally, the paper needs improvements in syntax for clarity and readability."
Some suggestions:
Line 30-31 Repharse : The duration of storage greatly influences the physicochemical properties and nutrient content of wheat, in addition to the environmental factors that are commonly reported.
LIne 69-72: I suggest to repharse: To address this issue, we conducted a long-term follow-up survey to investigate changes in the physicochemical properties and nutritional compositions of wheat during different storage periods. The findings of this study can provide essential insights into maintaining optimal wheat quality during practical storage.
Line189 “we performed to monitor” rephrase “we monitored”
Line 192-195 Need improvements to the syntax for clarity and readability.
Line 197: possible rephrasing: Throughout storage, wheat lipids undergo gradual hydrolyzation and oxidation, causing the fatty acid content in wheat to increase from 12.47 to 29.02mg/100g (Figure 1b). This increase in fatty acid content represents a significant deterioration in wheat quality.
Line 218 The extensibility?? You mean the swelling
Line 223-224 Rephrase The highly branched structure of amylopectin can absorb and hold water, which facilitates starch swelling. [28].
Delete lines 240-241.
Lines 242-249 the syntax needs improvement. Possible rephrasing: During wheat starch retrogradation, amylose, as a linear polymer, produces entangled cross-linking through hydrogen bonds to form a three-dimensional structure. These structures then accumulate to form crystals. In contrast, amylopectin has a dendritic branch structure with large intermolecular steric hindrance. As a result, the intermolecular movement or rearrangement through hydrogen bonds is greatly hindered, and the retrogradation rate is slowed down. The decreased amylose content inhibits gel formation during the retrogradation process, which leads to higher adhesive strength, as shown by the results.
Line 299-307 Table 4.
These results do not support any evolution of the amylose content over the two years of storage. Results in Table 4 are not commented nor explained
Lines 311-322 Based on my experience and the reliability of crystallinity measurement using X-ray diffraction, there was no significant difference between the 0 and 2 years of storage.
Line 341-342 The reduced content of amylose leads to the formation of double helix structure between the branched chains of amylopectin molecules, causing an increase in the order of starch granules.
Please be serious. In my opinion, the change in amylose content is not really shown by biochemical measurement and certainly cannot impact FTIR spectra."
Author Response
Thank you for all the careful and useful suggestions:
Reviewer: 1
1. Line 30-31 Repharse: The duration of storage greatly influences the physicochemical properties and nutrient content of wheat, in addition to the environmental factors that are commonly reported.
Author: Thank you. We have modified the sentence according to the suggestion and marked it out in the manuscript.
2.Line 69-72: I suggest to repharse: To address this issue, we conducted a long-term follow-up survey to investigate changes in the physicochemical properties and nutritional compositions of wheat during different storage periods. The findings of this study can provide essential insights into maintaining optimal wheat quality during practical storage.
Author: Thank you. We have repharsed the sentence according to the suggestion and marked it out in the manuscript.
3.Line189 “we performed to monitor” rephrase “we monitored”
Author: Thank you. We have changed it in the paper.
4.Line 192-195 Need improvements to the syntax for clarity and readability.
Author: Thank you. We have improved the syntax of Line 192-195.
5.Line 197: possible rephrasing: Throughout storage, wheat lipids undergo gradual hydrolyzation and oxidation, causing the fatty acid content in wheat to increase from 12.47 to 29.02mg/100g (Figure 1b). This increase in fatty acid content represents a significant deterioration in wheat quality.
Author: Thank you for pointing this out. We have modified the description in this paragraph.
6.Line 218 The extensibility?? You mean the swelling
Author: Thank you. Yes. It should be The extensibility. We have changed it in the paper.
7.Line 223-224 Rephrase The highly branched structure of amylopectin can absorb and hold water, which facilitates starch swelling. [28]
Author: Thank you. We have modified the sentence according to the suggestion.
8.Delete lines 240-241.
Author: Thank you. We have deleted the sentences.
9.Lines 242-249 the syntax needs improvement. Possible rephrasing: During wheat starch retrogradation, amylose, as a linear polymer, produces entangled cross-linking through hydrogen bonds to form a three-dimensional structure. These structures then accumulate to form crystals. In contrast, amylopectin has a dendritic branch structure with large intermolecular steric hindrance. As a result, the intermolecular movement or rearrangement through hydrogen bonds is greatly hindered, and the retrogradation rate is slowed down. The decreased amylose content inhibits gel formation during the retrogradation process, which leads to higher adhesive strength, as shown by the results.
Author: Thank you. We have We have improved the syntax and modified the sentences according to the suggestion.
10.Line 299-307 Table 4.
These results do not support any evolution of the amylose content over the two years of storage. Results in Table 4 are not commented nor explained
Author: Thank you. We have modified the description according to the suggestion.
11.Lines 311-322 Based on my experience and the reliability of crystallinity measurement using X-ray diffraction, there was no significant difference between the 0 and 2 years of storage.
Author: Thank you. We have changed the description according to the suggestion.
12.Line 341-342 The reduced content of amylose leads to the formation of double helix structure between the branched chains of amylopectin molecules, causing an increase in the order of starch granules.
Author: Thank you. We have modified the description according to the suggestion.
13.Please be serious. In my opinion, the change in amylose content is not really shown by biochemical measurement and certainly cannot impact FTIR spectra."
Author: Thank you for the useful and careful suggestion. We will pay attention to this and improve our design in the study.

Reviewer 2 Report
Points highlighted in the manuscript require clarification

Author Response
Thank you so much for the careful and useful suggestions:
Reviewer: 2
1.Line 89, specify which pretreatment is applied to the sample
Author: Thank you for the suggestion. As suggested by reviewers, we have added the pretreatment details in the paper.
2.Lines 297, Include the meaning of PV, TV, BD, FV, SB Units?
Author: Thank you. We have explained the meaning of these letters and added the Units in Table3.
3.Line 299, This increase is not statistically significant.
Author: Thank you. We have changed the description according to the suggestion.
4.Line309, Include the meaning of AAC. A, B1, B2, B3, ACL.
Author: Thank you for the suggestion. We have added the meaning of them in Table4.
5.Line342, Is the reduction in amylose significant enough to influence this change?
Author: Thank you. Sorry. It is a mistake. We have modified the description according to the suggestion.
6.Line370, The results of this study show that wheat should be stored less than one year to maintain its quality. - To review
Author: Thank you. Sorry. It is a mistake. We have modified the stored time according to the suggestion.
